# Effects of Dezocine on the Reduction of Emergence Delirium after Laparoscopic Surgery: A Retrospective Propensity Score-Matched Cohort Study

**DOI:** 10.3390/jpm13040590

**Published:** 2023-03-28

**Authors:** Lu Wang, Qiong Yi, Chunyan Ye, Ning Luo, E Wang

**Affiliations:** 1Department of Anesthesiology, Xiangya Hospital Central South University, Changsha 410008, China; 2National Clinical Research Center for Geriatric Disorders (Xiangya Hospital), Changsha 410008, China

**Keywords:** dezocine, emergence delirium, laparoscopy surgery, propensity score

## Abstract

In China, dezocine is commonly employed as a partial agonist of mu/kappa opioid receptors during anesthesia induction for surgical patients, yet evidence supporting its causal association with emergence delirium is limited. The objective of this investigation was to evaluate the impact of intravenous dezocine administered during anesthesia induction on emergence delirium. The retrospective studied existing data containing medical records of patients undergoing an elective laparoscopy procedure and the study was conducted with ethics-board approval. The primary outcome was the incidence of emergence delirium. Secondary outcomes included the VAS in the PACU and 24 h after surgery, the RASS score in the PACU, postoperative MMSE, hospital stay, and ICU stay. A total of 681 patients were analyzed, after being propensity score-matched, the dezocine and non-dezocine group each had 245 patients. Emergence delirium occurred in 26/245 (10.6%) of patients who received dezocine and 41/245 (16.7%) of patients did not receive dezocine. Patients on whom dezocine was used were associated with a significantly lower incidence of emergence delirium (absolute risk difference, −6.1%, 95% CI, −12% to −0.2%; relative risk [RR], 0.63; 95% CI, 0.18–0.74). All secondary outcome measures and adverse outcomes were not significantly different. The use of dezocine during anesthesia induction was associated with a decreased incidence of emergence delirium after elective laparoscopic surgeries.

## 1. Introduction

Delirium is a frequent complication that can arise after surgery and anesthesia. Postoperative delirium (POD) and emergence delirium (ED) are often distinguished in research, with POD evaluation typically commencing on postoperative day one, while ED assessment is conducted much earlier, often within the post-anesthesia care unit. ED is an acute confusion state following surgery and anesthesia; clinically, it may present as disorientation, hallucination, restlessness, or purposeless hyperactive physical behaviors. ED has been shown to be associated with increased healthcare cost [1], incidence of POD [2] and cognitive decline [3]. Until now, most of the research about ED has focused on specific age groups (preschoolers) and surgical categories (e.g., ENT, ophthalmology) which may be associated with a higher incidence of ED [4]. Therefore, there is no consensus on effective measures for ED prevention and treatment.

Although pre-clinical and clinical studies have implicated surgery and its associated stress and trauma as a potential source of systemic inflammation and subsequent neuroinflammation, little is currently known about the exact mechanisms that underlie emergence delirium (ED). This inflammation has been linked to functional brain connectivity impairment, gut wall injury and alterations in microbiome composition, all of which are believed to contribute to the development of neurological complications, including delirium [3]. Risk factors for ED, such as old age, major surgery, frailty and infection, have also been identified in these studies [3,5,6,7].

Dezocine is a mixed mu/kappa partial agonist and norepinephrine uptake inhibitor [8]. Dezocine provides effective pain relief in patients having inguinal hernia repair [9], open abdominal surgery [10], laparoscopic cholecystectomy and gynecological surgery [11,12,13]. Comparing with the commonly used mu receptor agonists such as fentanyl, dezocine is associated with less respiratory depression, constipation, psychological symptoms and abuse [8]. In our institution, dezocine was used during anesthesia induction at the discretion of the attending anesthesiologists. Patients treated with dezocine appeared to have less chance of having ED (anecdotal observation). Studies performed in pediatric surgical patients suggested that dezocine may reduce ED [14,15]. The effect of dezocine in adult surgical patients has been inadequately studied. Moreover, the previous studies failed in using methods to rigorously assess ED, e.g., the use of the Richmond Agitation-Sedation Scale (RASS) and the Confusion Assessment Method for the Intensive Care Unit (CAM-ICU).

In this study, we hypothesize that dezocine used during anesthesia induction is associated with a reduced ED incidence in adult surgical patients. This retrospective analysis was based on patients who underwent elective laparoscopic surgery from 2017 to 2019 in our hospital.

## 2. Materials and Methods

### 2.1. Study Design

This retrospective study was approved by Xiangya Hospital’s Institutional Review Board on 25 July 2022 (IRB #202207168), with patient consents waived. The report of this study followed the Strengthening the Reporting of Observational Studies in Epidemiology (STROBE) guidelines.

### 2.2. Setting

The patients included in this study received surgery and care in a tertiary teaching hospital, i.e., Xiangya Hospital, which is affiliated with Central South University and located in Changsha in China.

### 2.3. Participants

This retrospective study was based on patients who received elective laparoscopy surgeries from October 2017 to July 2019. Patients were included in this study if they had elective laparoscopic procedures. Patients were excluded from this study if they had an American Society of Anesthesiologists (ASA) physical status ≥ IV, preoperative cognitive impairment (defined as the Mini-Mental State Examination (MMSE) score ≤ 23), and medical records with missing perioperative data (especially objective ED assessment which described in the following section).

### 2.4. Variables

The primary outcome was the incidence of emergence delirium, which was assessed using the Confusion Assessment Method for the Intensive Care Unit (CAM-ICU) scale 20 min after tracheal extubation. The CAM-ICU scale was used if the Richmond Agitation Sedation Scale (RASS) score was ≥−3 [16]. If the CAM-ICU assessment was positive, emergence delirium was diagnosed [17]. If the RASS score was −4 or −5, the assessment was postponed until the RASS score was ≥−3.

The exposure was the use of dezocine during the anesthesia induction which was obtained from the case management system in the medical record room of the Xiangya Hospital, which is affiliated with Central South University.

The potential confounders in this study included age, gender, BMI, Charlson Score, years of education, preoperative MMSE score, type of surgery, lesions, operation time and anesthesia time, which were regarded as risk factor in other studies and from clinical evidence-based experience.

### 2.5. Data Sources/Measurement

The data were derived from the electronic medical records kept in Xiangya hospital. The ED data was based on the PACU records kept by the Department of Anesthesiology in Xiangya Hospital. The PACU records correspond to the research project (ChiCRT2000031201, NCT03330236) and other preliminary research projects.

### 2.6. Bias

The selection bias was minimal because of the inclusive nature of recruitment. As the ED assessment of all our patients were evaluated in PACU settings, which was measured using the same methods by professional staff. Therefore, the information bias was minimal. Propensity score matching was used to balance the preoperative characteristics between the two cohorts.

### 2.7. Study Size

No statistical power calculation was conducted before the study because we planned to include all patients who met the selection criteria. The sample size was based on the available cases.

### 2.8. Quantitative Variables

Most quantitative data were obtained from either the monitors or the electronic medical records without modification, including demographic characteristics, laboratory results, vital signs and drug doses. We removed data outside of the 0.5 to 99.5 percentile range for vital signs, considering that some of these measurements could be artifacts or outliers.

### 2.9. Statistical Analysis

Propensity score matching was performed per age, gender, BMI, Charlson Score, years of education, preoperative MMSE score, type of surgery, lesions, operation time and anesthesia time between patients who received and did not receive dezocine during anesthesia induction. Nearest neighbor 1:1 propensity score matching without replacement using a logit model was performed on these data and matching standardized differences (the difference in means or proportions divided by the pooled standard deviation) for each covariate was used to assess the performance of propensity score matching. The weighted matched standardized difference between groups was control less than 0.1 [17,18].

Data were expressed as mean ± SD, median (IQR) and number (percentage) and analyzed with unpaired *t* test or the Mann–Whitney U test where appropriate for continuous variables, otherwise, a chi-square was used. The relative risks (RRs) of outcomes were estimated for patients with and without dezocine treatment. All data were analyzed with SPSS software (Version 26.0, IBM Inc., Armonk, NY, USA).

## 3. Results

### 3.1. Participants

The flow chart for patient inclusion is shown in the Figure 1. Of 2344 patients with data, 1573 were excluded due to medical records with missing perioperative data including incomplete postoperative delirium assessment. The reasons for the missing delirium data might include missing the assess time, patients’ refusal, stupor or nonresponsive, or postoperative tracheal intubation. An additional 66 patients were excluded due to preoperative cognitive impairment and 24 subjects were ASA Ⅳ, the final analysis included 681 patients (Table 1).

### 3.2. Descriptive Data

The database interrogation yielded a total of 681 patients with emergence delirium. Patients’ baseline characteristics are presented in Table 2. Univariate logistic regression analyses showed that lower education year (OR, 0.826; 95% CI, 0.883–0.944; *p* < 0.001), lower BMI (OR, 0.916; 95% CI, 0.844–0.993; *p* = 0.034) and longer operation time (OR, 1.005; 95% CI, 1.003–1.007; *p* < 0.001) were associated with an increased risk for ED. Contrarily, malignant lesions (not shown), Charlson score, age and preoperative MMSE score was not associated with the risk for developing ED (Figure 2).

Before propensity score matching, standardized differences in preoperative characteristics ranged between −0.224 and 0.286. To further investigate this imbalance, we illustrate histogram of the distribution of the propensity score for both groups before and after propensity matching. Figure 3 (left) presents histograms of unbalanced propensity score distribution for both groups before propensity matching. Figure 3 (right) presents histograms of balanced propensity score distribution for both groups after the propensity matching.

After propensity score matching, all standardized differences were less than |0.10|, indicating that in the propensity matching, 245 patients, who received dezocine during anesthesia induction, were matched to 245 those patients who did not receive dezocine. The groups were comparable in the baseline variables (Table 3) were comparable between two groups.

### 3.3. Emergence Delirium Incidence

The dezocine group had a lower incidence of emergence delirium than that of the non-dezocine group (10.6% vs. 16.7%; absolute risk difference, −6.1%; 95% CI, −12% to −0.2%; RR, 0.63; 95% CI, 0.18–0.74). (Table 4 and Appendix A).

### 3.4. Other Outcomes

There were no differences of VAS scores in the PACU or 24 h after operation between the two groups (Table 4). No significant difference was found regarding nausea and vomiting, acute kidney injury, MMSE during hospital stay, the length of ICU and hospital stay (Table 4).

No patients were given benzodiazepines or anticholinergics before surgery. All other aspects including PACU stay time, extubation time, intraoperative fluid load (crystalloid and colloid), blood transfusion, intraoperative hypotension, use of vasopressor drugs, intraoperative medications including benzodiazepines and atropine, the dose of medication administered for anesthesia maintenance propofol, remifentanil, sufentanyl or sevoflurane for anesthesia maintenance and postoperative analgesia were not significantly different between the two groups (Table 5).

## 4. Discussion

### 4.1. Key Results

In this study, this retrospective cohort study examined 2344 patients who received elective laparoscopy surgeries from October 2017 to July 2019, and the final analysis included 681 patients. Analyses based on propensity score matching suggested that intravenous dezocine administered during the induction of anesthesia was associated with a lower risk of emergence delirium.

### 4.2. Interpretation

ED is a disturbance of consciousness during recovery from general anesthesia, and includes hallucinations, delusions and confusion. Many terms have been used interchangeably with ED, such as emergence agitation, postoperative delirium, paradoxical excitement [19] and postanesthetic delirium [20]. According to the DSM-V [21], delirium is “a disturbance in attention and awareness based on the following criteria: disturbance in the level of awareness and reduced ability to direct, focus, sustain, and shift attention; a change in cognition associated with evidence from the patient history, physical exam, or laboratory findings that the disturbance is caused by the direct physiologic consequences of a general medical condition; this disturbance develops over a short period of time and tends to fluctuate in severity during the course of the day”.

Notably, given the observed ED prevalence of 11.7% among patients in the study, the relatively low incidence in our study is in contrast to previous studies [22,23,24]. This may be attributable to several reasons. First, intravenous anesthesia (TIVA) was applied in the majority of patients in the study, and different anesthetic techniques may lead to different incidences of ED, as suggested by a previous comparative study that found that ED occurred in 6.9% of patients receiving propofol anesthesia and in 26.7% of patients receiving sevoflurane anesthesia, the latter agents promote early arousal, which contributes to ED [25]. Second, the use of benzodiazepines and anticholinergics was not recommended before surgery in our hospital, as these drugs may be included in the increased risk factor list for delirium [26].

Regarding the clinical features of ED and in line with the published literature, our analyses revealed that longer operation time, less education year and lower BMI were significant predictors of ED development, although age is a well-recognized risk factor for delirium, an age greater than 65 years bears an increased risk for delirium [27], we found no evidence that age was related to ED.

Currently, numerous latent interventions for managing ED have been studied, including medications and behavioral interventions. Among them, dexmedetomidine, a highly selective α2 adrenergic receptor agonist, has been suggested as a potential prophylactic intervention against sevoflurane-associated ED. It provides sedation and anxiolysis by acting on these receptors in the locus coeruleus of the pons. It also exerts dose-dependent analgesic effects through binding to α2-receptors in the dorsal horn and the supra-spinal sites [28,29,30,31]. However, there are little reports yet to dezocine during surgery use association with postoperative delirium. As an analgesic, the analgesic effect of dezocine is approximately equipotent to that of morphine. It has a minor ceiling effect of respiratory depression, and its sedative effect is well tolerated with in a single-dose use. The use of dezocine in Chinese cancer patients showed that there was no significant difference between dezocine and morphine in terms of antinociceptive efficacy, and the rate of adverse reactions reported for dezocine was less than that for morphine [32]. Additionally, dezocine possesses norepinephrine uptake inhibitory activity, which could synergize with mu agonists in the case of acute pain treatment and possibly endow the drug with good pain relief in neuropathic pain conditions [33]. Previously, we and others found that that the use of dezocine during the induction of anesthesia suppressed the occurrence of sufentanyl-induced coughing, including RCTs [34,35,36] and meta-analyses [37,38]. For these reasons, owing to the preference of anesthesiologists, in our practice, some patients received approximately 0.1–0.15 mg/kg of dezocine during induction of anesthesia before sufentanyl was given.

In our study, 26 patients in the dezocine group and 41 patients in the non-dezocine group developed emergence delirium. The matched pairs were identified using the 1:1 nearest neighbor method with a 0.02 caliper, so the balance between groups was conducive to detecting differences. The quality of delirium assessment was adjudicated by an external expert panel without purposely knowing whether the patient had received dezocine. Therefore, it was less likely that ascertainment bias accounted for the observed association between dezocine and delirium in the present study.

The underlying mechanisms are unknown. Pain is one of inducers of postoperative delirium [39], but our current study showed that dezocine did not provide an extra postoperative relief during the PACU and at 24 h after the operation. Nonetheless, we considered the lack of statistically significant pain score results to make the conclusion more intuitive because inadequate pain control remains a potential cause of or contributor to the incidence of ED after brief surgical procedures [39]. If the pain score had been imbalanced between the two groups, it would have contributed to the presence of ED.

We further analyzed the main anesthetic amount which including propofol, remifentanil, sufentanyl and the use of sevoflurane in the two groups, and the difference was insignificant. Studies showed that TIVA with propofol is associated with a much lower incidence of ED than sevoflurane-based anesthetic techniques [40]. Moreso, the insignificant drugs dosages, to some extent, might suggest that insignificant difference of depth of anesthesia between the cohorts, which was in line with similar extubation time between the two cohorts.

The extubation time between the cohort were similar. The study conducted by Bong et al. [31] concluded that recovery time is an important predictor of ED, with every minute increase in wake-up time reducing the odds of delirium by 7%. To our knowledge, ED is generally self-limiting and can be relieved after approximately 5–15 min in most cases. The results of the current study also showed that PACU length of stay after extubation was similar between the two groups. Furthermore, there was no difference in the length of hospital stay.

We paid attention to intraoperative hemodynamics since low blood pressure during surgery can lead to inadequate oxygen supply to the brain, potentially causing symptoms of brain dysfunction such as delirium. However, our analysis did not find any significant differences in the incidence of hypotension or use of vasopressor drugs. We speculated that due to dezocine having a strong analgesic effect due to its unique pharmacological action, its side effects are very slight. Study [41] showed that dezocine could reduce the postoperative MAP and HR fluctuation, effectively reduce the effect of stress factors and help maintain the stability of the brain environment. However, our study has demonstrated that dezocine is capable of preserving hemodynamic stability. Specifically, a single dose of dezocine does not appear to impose an additional burden on hemodynamics.

Finally, we tried to interpret the mechanism of the reduction in ED. It is likely that the pharmacological effects of dezocine may contribute to the incidence of delirium decrease. Liu et al. [42] found two novel molecular targets: norepinephrine transporter NET and serotonin transporter SERT of dezocine. As a kappa antagonist, dezocine shares a binding site with certain clinical antidepressant drugs, whereas the kappa opioid receptors associated with NET and SERT are an important target for depressant action [32]. Taken together, dezocine was proposed to an alternative medication to treat depression [43,44]. Interestingly, perioperative depression was reported to be associated with worse surgical outcomes including neurological complications including delirium, in various surgeries including colorectal [45], spine [46] and cardiac surgery [47]. Therefore, anti-depression effects of dezocine may be responsible for the decreased emergence delirium incidence although this need to be studied further.

### 4.3. Limitations

There are several potential limitations in our study. First, although we adjusted for several potential confounding factors using the propensity score-matching method, there may be other factors that cannot be controlled, which may affect the incidence of delirium. Second, we focused on measuring emergence delirium once in the PACU. As a result, we may have missed some important characteristics of ED. Thirdly, the very low POD incidence in our study is in contrast to previous studies, as a result of anesthetic mode and the age of the patient involved in the study, so the effect of dezocine on POD remains ambiguous. This may be attributable to several reasons. Finally, our studies focused on patients who underwent elective laparoscopy surgery, and the results may not be directly generalized to patients undergoing emergence or other types of surgery. To shed light on the effect of dezocine on delirium, we still need to carry out more prospective studies to make a broader generalization about whether dezocine can be used without concern for emergence delirium, even for postoperative delirium.

## 5. Conclusions

Our study indicated that dezocine use during anesthesia induction is associated with a lower incidence of emergence delirium during PACU stay. Whether dezocine reduces emergence delirium requires further study.

## Figures and Tables

**Figure 1 jpm-13-00590-f001:**
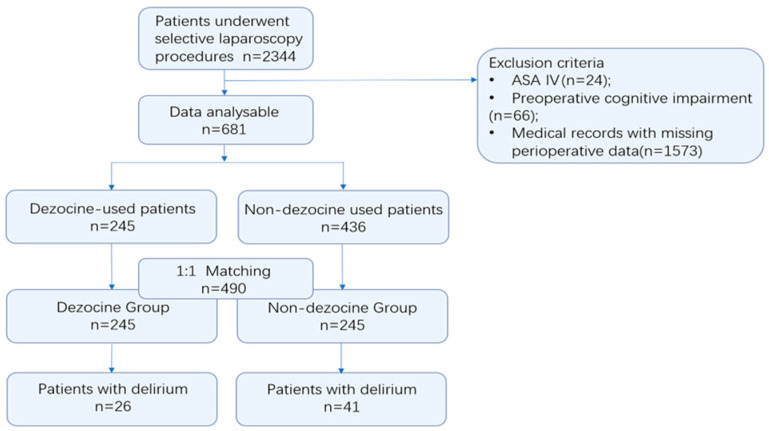
Participants’ recruiting flow chart. The inclusion and exclusion criteria for the propensity score-matched cohort are presented.

**Figure 2 jpm-13-00590-f002:**
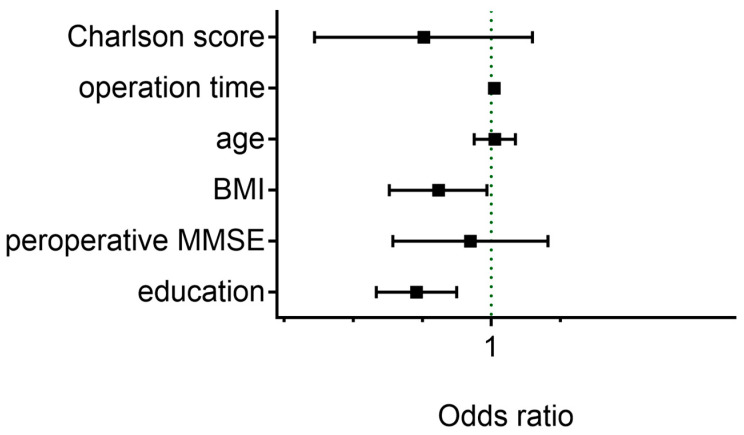
Odds ratios for emergence delirium (ED) with 95% confidence intervals.

**Figure 3 jpm-13-00590-f003:**
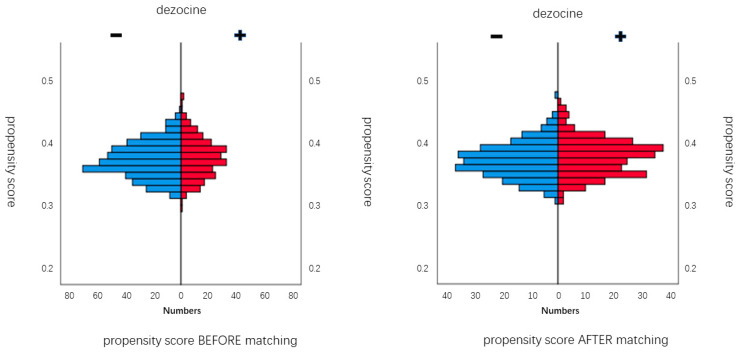
Histograms of propensity score distribution before and after propensity score matching. Distribution of the propensity scores before and after matching for the dezocine group (+) and the non-dezocine group (−). (**Left panel**) presents histograms of unbalanced propensity score distribution in both groups before propensity matching. (**Right panel**) presents histograms of balanced propensity score distribution in both groups after propensity matching.

**Table 1 jpm-13-00590-t001:** Baseline characteristics (n = 681).

Categories	Mean ± SD and Median [IQR] and Number of Patients (%)
Demographics	
Age, year	60 ± 7.3
Gender (Female)	371 (54.3%)
BMI, kg/m^2^	23.5 ± 3.2
Charlson score	0 (0–2)
Education, year	9 (6–12)
Preoperative MMSE score	29 (27–30)
Type of surgery	
Gastrointestinal surgery	237 (34.8%)
Urogenital surgery	246 (36.1%)
Hepatobiliary surgery	137 (20.1%)
Others	61 (9%)
Lesions (malignant)	382 (56.1%)
Operation time ^a^, min	156 ± 84
Anesthesia time ^b^, min	226 ± 98

Abbreviations: BMI, body mass index; SD, standard deviation; IQR, interquartile range; MMSE, mini-mental status exam. ^a^ The operation time started with skin incision and ended with wound closure. ^b^ The anesthesia time started with anesthesia induction and ended with tracheal extubation.

**Table 2 jpm-13-00590-t002:** Prevalence, risk factors and clinical characteristics of ED.

Patient Characteristics	Patient with ED *n* = 80	Patient without ED *n* = 601	*p* Value
Age, mean ± SD, Year	61.4 ± 7.7	59.5 ± 7.0	0.047 ^a,^*
Female, *n* (%)	32(40%)	294 (48.9%)	0.153 ^b^
BMI, mean ± SD, kg/m^2^	22.7 ± 3.1	23.8 ± 3.0	0.001 ^a,^*
Charlson score, median (IQR)	2 (0–2)	0 (0–2)	0.039 ^a,^*
Education, median (IQR), Year	8 (5–9)	9 (7–12)	<0.001 ^a,^*
Preoperative MMSE score, median (IQR)	28 (26–29)	29 (28–30)	0.026 ^a,^*
Lesion (malignant), *n* (%)	55 (68.8%)	327 (54.4%)	0.016 ^b,^*
Operation time, mean ± SD, min	197 ± 84	151 ± 82	<0.001 ^a,^*
Anesthesia time, mean ± SD, min	279 ± 96	219 ± 96	<0.001 ^a,^*

^a^ Mann–Whitney U test. ^b^ χ^2^ test. * Denotes significance at *p* < 0.05.

**Table 3 jpm-13-00590-t003:** Comparisons between cohort preoperative characteristics before and after propensity score matching.

Variable	Before Propensity Score Matching			After Propensity Score Matching		
	Dezocine Group (*n* = 245)	Non-Dezocine Group (*n* = 436)	SMD ^a^	Dezocine Group (*n* = 245)	Non-Dezocine Group (*n* = 245)	SMD
Age, mean ± SD, year	61.6 ± 7.3	59.2 ± 7.1	0.237	61.6 ± 7.3	60.6 ± 7.4	0.096
Female, *n* (%)	102 (41.6%)	208 (47.7%)	−0.082	102 (41.6%)	108 (44.1%)	0.029
BMI, mean ± SD, kg/m^2^	23.5 ± 3.3	23.4 ± 3.1	0.016	23.5 ± 3.3	23.0 ± 3.0	0.090
Charlson score, median (IQR)	2 (0–2)	0 (0–2)	−0.224	2 (0–2)	2 (0–2)	0.004
Education, median (IQR), Year	9 (6–12)	9 (7–12)	0.045	9 (6–12)	9 (6–12)	0.061
Preoperative MMSE score, median (IQR)	28 (26–29)	29 (28–30)	0.286	28 (26–29)	28 (27–29)	−0.014
Type of surgery						
Gastrointestinal surgery, *n* (%)	99 (40.4%)	138 (31.6%)	0.038	99 (40.4%)	107 (43.7%)	−0.007
Urogenital surgery, *n* (%)	90 (59.6%)	156 (35.8%)		90 (59.6%)	81 (56.3%)	
Hepatobiliary surgery, *n* (%)	38 (15.5%)	99 (22.7%)		38 (15.5%)	37 (15.1%)	
Others, *n* (%)	18 (0.07%)	43 (0.1%)		18 (0.07%)	20 (0.08%)	
Malignant, *n* (%)	163 (66.5%)	219 (50.2%)	0.23	163 (66.5%)	159 (64.9%)	0.02
Operation time ^b^, mean ± SD, min	172 ± 85	148 ± 82	0.202	172 ± 85	182 ± 87	−0.082
Anesthesia time ^c^, mean ± SD, min	242 ± 99	217 ± 96	0.182	242 ± 99	254 ± 105	−0.082

Abbreviations: BMI, body mass index; SD, standard deviation; IQR, interquartile range; MMSE, mini-mental status exam. ^a^ SDM: standardized mean difference, the difference in means or proportions divided by the pooled standard deviation. ^b^ The operation time started with skin incision and ended with wound closure. ^c^ The anesthesia time started with anesthesia induction and ended with tracheal extubation.

**Table 4 jpm-13-00590-t004:** Outcomes in the propensity score—matched cohorts.

Outcomes	Dezocine Group (*n* = 245)	Non-Dezocine Group (*n* = 245)	*p* Value
Primary outcome			
Emergency delirium, *n* (%)	26 (10.6%)	41 (16.7%)	0.049
Secondary outcome			
VAS ^a^, In PACU, median (IQR)	3 (0–5)	2 (0.5–4)	0.34
VAS ^a^, 24 h after surgery, median (IQR)	2 (1–4)	2 (1–4)	0.35
RASS ^b^, In PACU, median (IQR)	0 (−1–0)	0 (−1–0)	0.226
Nausea and vomiting, *n* (%)	59 (24.1%)	63 (25.7%)	0.67
Acute kidney injury, *n* (%)	14 (5.7%)	20 (8.2%)	0.286
MMSE, before discharge, median (IQR)	28 (26–29)	28 (26–29)	0.93
Hospital stay, median (IQR), days	6 (3–8)	6 (3–8)	0.76
Intensive care unit stay, *n* (%)	4 (0.8%)	7 (1.4%)	0.360

Abbreviations: VAS, visual analogue scale; RASS, Richmond Agitation Sedation Scale; MMSE, mini-mental status exam; IQR, interquartile range. ^a^ Pain severity was assessed using the visual analogue scale (VAS), which is a reliable instrument for the measurement of pain intensity self-reported by the patient (ranging from 0 to 10, with higher scores indicating more pain). ^b^ The Richmond Agitation Sedation Scale is comprised of a 10-point range, spanning from −5 for an unresponsive patient who does not react to any stimuli, to +4 for a highly agitated and combative patient.

**Table 5 jpm-13-00590-t005:** Perioperative characteristics of the two study cohorts.

	Dezocine Group (*n* = 245)	Non-Dezocine Group (*n* = 245)	*p* Value
Intraoperative Data			
PACU stay, median (IQR), min	80 (60–120)	80 (60–110)	0.291
Extubation time, median (IQR), min	35 (25–50)	35 (25–50)	0.464
Crystalloid, median (IQR), mL	1100 (750–1600)	1100 (750–1600)	0.514
Colloid, median (IQR), mL	500 (0–500)	500 (0–500)	0.351
Blood transfusion, *n* (%)	9 (0.037%)	18 (0.073%)	0.075
Intraoperative hemodynamic			
Hypotension, *n* (%)	100 (40.8%)	86 (35%)	0.456
Vasopressor drugs, *n* (%)	71 (30.0%)	66 (27.0%)	0.920
Intraoperative Medications			
Benzodiazepines, *n* (%)	3 (0.122%)	3 (0.122%)	1.000
Atropine, *n* (%)	44 (18.0%)	52 (21.2%)	0.735
Medication administered for anesthesia maintenance			
Propofol, median (IQR), mg/kg/min	0.14 (0.1–0.2)	0.15 (0.9–0.2)	0.314
Remifentanil, median (IQR), µg/kg/min	0.18 (0.13–0.26)	0.19 (0.13–0.29)	0.318
Sufentanyl, median (IQR), µg/min	0.39 (0.35–0.43)	0.43 (0.35–0.43)	0.883
Sevoflurane, *n* (%)	1 (0.004%)	3 (0.122%)	0.623
Postoperative analgesia			
PCA pumps, *n* (%)	139 (56.7%)	143 (58.4%)	0.829
multimodal analgesia, *n* (%)	6 (0.024%)	5 (0.02%)	1.000

Abbreviations: PACU, post anesthesia care unit; IQR, interquartile range.

## Data Availability

The data presented in this study are available on request from the corresponding author.

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
