# Peer review of "Effects of Dezocine on the Reduction of Emergence Delirium after Laparoscopic Surgery: A Retrospective Propensity Score-Matched Cohort Study"

_jpm, 2023, doi:10.3390/jpm13040590_

Round 1

Reviewer 1 Report

The authors have addressed all the comments that were made in the initial review. They present the results of an interesting and well designed study about a very common and severe complication (postoperative delirium) in ICU, a population with high incidence of POD. I would like to congratulate the authors for their effort and this very interesting study.

Author Response

 Thank you for your encouraging remarks and valuable comments. Our study indicated that dezocine use during anesthesia induction is associated with a lower incidence of delirium during PACU stay. But we remained convinced the evidence was still limited because the study was retrospective, as a result, whether dezocine reduces emergence delirium requires further study.

The authors have addressed all the comments that were made in the initial review. They present the results of an interesting and well designed study about a very common and severe complication (postoperative delirium) in ICU, a population with high incidence of POD. I would like to congratulate the authors for their effort and this very interesting study.

Response: Thank you for your encouraging remarks and valuable comments. Our study indicated that dezocine use during anesthesia induction is associated with a lower incidence of delirium during PACU stay. But we remained convinced the evidence was still limited because the study was retrospective, as a result, whether dezocine reduces emergence delirium requires further study.

Reviewer 2 Report

I would prefer to see the distribution of the propensity score in a graph in the two cohorts. I would ask you at added it

Author Response

   Thank you for your valuable suggestion, propensity score in the two cohorts will be added in the manuscript.

Reviewer 3 Report

In this intresting study authors are evaluating the preventive properties of dezocin regarding the emergence delirium.

The manuscript needs linguistic editing prior to final evaluation. At some points emergence is written as emergency and there are further typos, grammatical and structural errors in the text. 

Study design seems a little vague while reading the manuscript despite the quite specific manuscript title. The terminology used at some points corresponds to a prospective study in the description rather than a retrospective one. (The part regarding patient approach – bias section : "All potentially eligible patients during the study period were approached for this study"). Do you mean that you retrospectively approached these patients to obtain informed consent ? Was it really needed ? Were not the data anonymized and coded ?  Please rephrase to avoid reader doubts.

Page 7, interpretation section, last line:  emergence should be added before delirium as this was your specific outcome and not post-operative delirium in general. 

Please add some info regarding preoperative cognitive assessment . Is it standard of care in your organisation to evaluate all patients prior to this using MMSE?

Regarding the age groups included, it is mentioned that age is an identified risk factor, could you please include some further info/results regarding this correlation? Have you checked if your ED+ patients belong to the edges of your age group as regarding POD an exponential increase of incidence has been shown in  ages >65. A separate age-based cohort analysis could be included if possible.     

Author Response

  We deeply appreciated the time and efforts you have spent in reviewing our manuscript. Your comments are really thoughtful and helpful. Thus we revisesd the manuscript, following your comments. Specific responses to  your comments will be attached after the note.

Response to Reviewer 3 Comments

  1. In this intresting study authors are evaluating the preventive properties of dezocin regarding the emergence delirium. The manuscript needs linguistic editing prior to final evaluation. At some points emergence is written as emergency and there are further typos, grammatical and structural errors in the text.

Response:The statements of “emergency” were corrected as “emergence”.

  1. Study design seems a little vague while reading the manuscript despite the quite specific manuscript title. The terminology used at some points corresponds to a prospective study in the description rather than a retrospective one. (The part regarding patient approach – bias section : "All potentially eligible patients during the study period were approached for this study"). Do you mean that you retrospectively approached these patients to obtain informed consent ? Was it really needed ? Were not the data anonymized and coded ? Please rephrase to avoid reader doubts.

Response:Following the suggestion of the referees, we have rephrase this part. Now it reads:” The selection bias was minimal because of the inclusive nature of recruitment. As the ED assessment of all our patients were evaluated in PACU settings, which was measured using the same methods by professional staff. Therefore, the information bias was minimal. Propensity score matching was used to balance the preoperative charac-teristics between the two cohorts.”(Line114-118). As a retrospective study, we approved by Xiangya Hospital’s Institutional Review Board on July 25, 2022 (IRB #202207168), with patient consents waived, and the data was anonymized and coded due to data protection and privacy regulations.

  1. Page 7, interpretation section, last line: emergence should be added before delirium as this was your specific outcome and not post-operative delirium in general.

Response:Following the suggestion of the referees, ”emergence” was added before delirium which was much more accurate.

  1. Please add some info regarding preoperative cognitive assessment . Is it standard of care in your organisation to evaluate all patients prior to this using MMSE?

Response:We are very sorry for our negligence of providing detailed information regarding preoperative cognitive assessment. The MMSE may serve as a screening tool for delirium, even though it was originally designed to assess cognitive impairment, so studies invloved delirium or POCD would evaluate MMSE score. Although MMSE was not a standard of care in our organisation, but patients involved in our study must have ED valuation in the PACU records which correspond to the research project (ChiCRT2000031201, NCT03330236) and other preliminary research projects, and therefore these patients at the same time had preoperative and postoperative MMSE score in the records.

  1. Regarding the age groups included, it is mentioned that age is an identified risk factor, could you please include some further info/results regarding this correlation? Have you checked if your ED+ patients belong to the edges of your age group as regarding POD an exponential increase of incidence has been shown in ages >65. A separate age-based cohort analysis could be included if possible.

Response:That was an excellent suggestion. Age is a recognized risk factor of delirium, and besides some other factors including gender, education , type of operation , operation duration and so on, which sugested in many studies might be related to delirium. patients’ characteristics between cohort with/without emergence delirium was analysed in Table2 and Figure2. We found that after univariate logistic regression analyses, lower education year (OR, 0.826; 95% CI, 0.883–0.944; p < 0.001), lower BMI(OR, 0.916; 95% CI, 0.844–0.993; p =0.034), and longer operation time(OR, 1.005; 95% CI, 1.003–1.007; p < 0.001) were associated with an increased risk for emergence delirium (Line165-Line170). Based on the results so far, Compared with postoperative delirium, age might not be such a prominent risk factor as far as emergence delirium was concerned. But that conclusion remained uncertain in view of the  retrospective and small sample study.